# Hardware-Aware Parallel Prompt Decoding for Memory-Efficient Acceleration of LLM Inference

## Abstract

The auto-regressive decoding of Large Language Models (LLMs) results in significant overheads in their hardware performance. While recent research has investigated various speculative decoding techniques for multi-token generation, these efforts have primarily focused on improving processing speed such as throughput. Crucially, they often neglect other metrics essential for real-life deployments, such as memory consumption and training cost. To overcome these limitations, we propose a novel parallel prompt decoding that requires only 0.0002% trainable parameters, enabling efficient training on a single A100-40GB GPU in just 16 hours. Inspired by the human natural language generation process, *PPD* approximates outputs generated at future timesteps in parallel by using multiple prompt tokens. This approach partially recovers the missing conditional dependency information necessary for multi-token generation, resulting in up to a 28% higher acceptance rate for long-range predictions. Furthermore, we present a hardware-aware two-stage tree pruning algorithm that adaptively optimizes this decoding scheme to fully leverage the computational capacities on different GPUs. Through extensive experiments across LLMs ranging from MobileLlama to Vicuna-13B on a wide range of benchmarks, our approach demonstrates up to 2.49× speedup and maintains a minimal runtime memory overhead of just 0.0004%. More importantly, our parallel prompt decoding can serve as an orthogonal optimization for synergistic integration with existing speculative decoding, showing up to 1.22× further speed improvement. Our code will be open-sourced upon acceptance of the paper.

## 1 Introduction

The recent advances in large language models (LLMs) are increasingly shaping and influencing a wide range of AI applications. However, autoregressive generation, the de facto approach employed in LLM inference, suffers from inadequate hardware performance due to its inherent sequential nature (Stern et al., 2018). Speculative decoding (Leviathan et al., 2023; Chen et al., 2023; Kim et al., 2024), an emerging acceleration technique, employs a guess-and-verify framework for LLM inference, where a smaller draft model first predicts multiple tokens sequentially and then the original

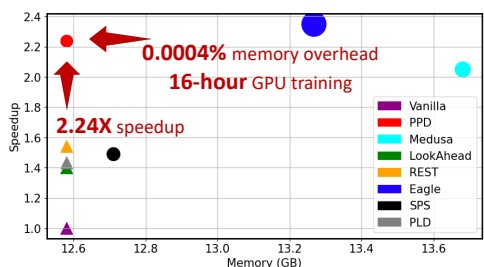

Figure 1: Comparison of memory, speedup, and training cost on MT-Bench with Vicuna-7B. Circle diameter shows training GPU hours.

LLM verifies them in parallel. Despite its potential, the effectiveness of speculative decoding is limited by the complexity and cost of training a draft model capable of consistently achieving high acceptance rates across diverse base models and datasets. Additionally, the extra runtime memory overhead for executing draft models poses a significant barrier to the broader adoption of speculative decoding, particularly in edge and mobile environments where memory capacity is limited. Considering the growing need for user privacy and personalization, deploying LLMs on devices urges a more memory- and cost-efficient solution for accelerating LLM inference. Recent efforts have explored the possibility of generating multiple tokens in parallel without relying on a separate transformer

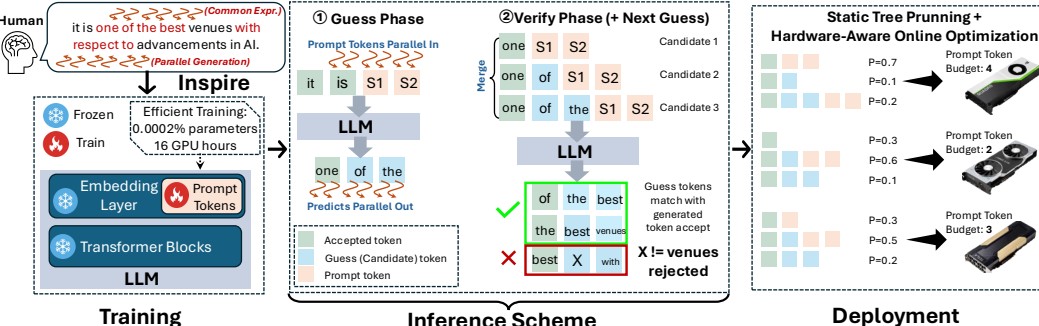

Figure 2: Overview of *PPD*. The left section shows the location of trainable parameters and the middle section displays the combined **guess-and-verify** process during inference. The "prompt token" denotes the special token with separately trained embeddings to perform parallel prediction.

draft model (Santilli et al., 2023). Approaches such as inserting additional decoding heads (Cai et al., 2024) and retrieving frequently used tokens (He et al., 2023) are employed to enhance performance. However, these methods either aggressively assume conditional independence among the tokens generated in a single step (Cai et al., 2024; He et al., 2023), or use placeholder tokens (*e.g.*, [PAD] token) that do not convey enough contextual information (Santilli et al., 2023). Therefore, they often suffer from low acceptance rates or degradation in output quality due to the lack of sufficient conditional information during inference.

To alleviate the complexity and overhead associated with the use of draft models while maintaining a high acceptance rate, we propose *Parallel Prompt Decoding* (*PPD*), a novel architecture-agnostic and memory-efficient framework that adopts prompt tuning for non-autoregressive LLM inference. Inspired by the human natural language generation process where continuous words like common expressions and phrases are produced simultaneously, *PPD* introduces the use of prompt tokens, the meticulously trained embeddings, for multi-token prediction. Specifically, these trained prompt tokens are appended to the original input sequence in parallel, enabling the concurrent generation of multiple output tokens in a single forward pass. The key intuition of *PPD* lies in the observation that if trained properly, prompt tokens appended to the input can approximate tokens generated at future timesteps, thereby partially recovering the missing conditional dependency information for multi-token generation. By strategically positioning trained prompt tokens, *PPD* achieves up to a 28% higher acceptance rate when predicting long-range tokens. To further increase the token acceptance rate, we generate multiple candidate continuations with each prompt token and use them in combination with a customized tree attention mask to minimize the computation and memory overhead. The capability of *PPD* to use low-cost prompt tokens for accurate multi-token prediction forms the foundation for accelerating LLM inference. As shown in Figure 1, *PPD* achieves a comparable speedup to the state-of-the-art speculative decoding approaches with negligible memory overhead and reduced training cost. Moreover, to facilitate the optimized implementation of *PPD* across different hardware platforms, we propose a hardware-aware two-stage tree pruning technique that adaptively refines the prompt structure during runtime based on the computational resources available on the specific hardware.

To demonstrate the effectiveness of our approach, we evaluate *PPD* on MobileLLaMA (Chu et al., 2023), Vicuna-7b and Vicuna-13b (Chiang et al., 2023). Running on a single GPU using the A100-40GB and RTX 4090, our method achieves a speedup ratio for inference from **2.12×** to **2.49×** across a diverse range of popular datasets including MT-Bench, HumanEval, and GSM8K. Our experiments demonstrate that *PPD* not only achieves comparable throughput to the state-of-the-art speculative decoding method, but it also manages this with significantly fewer trainable parameters—specifically, **0.0002%** of trainable parameters—and incurs only a minimal memory overhead (**0.0004%**), showcasing that *PPD* is remarkably cost- and memory-efficient. The training of prompt tokens can be completed in **16 hours** using one A100 GPU, **8 hours** using four GeForce RTX 3090 GPUs, compared to the 1-2 days on four A100 GPUs required for Eagle (Li et al., 2024a). Furthermore, since *PPD* does not require the modification of the original LLM or the addition of

extra networks, it is highly adaptable and orthogonal to other decoding techniques. For instance, it can be effectively combined with a draft model to further reduce inference latency.

Our contributions are summarized as follows:

- A novel *Parallel Prompt Decoding* (*PPD*) that adopts cost-effective prompt tokens for non-autoregressive LLM inference, achieving a high acceptance rate for long-distance token prediction with preserved output quality.

- A hardware-aware two-stage tree pruning technique that adaptively optimizes the prompt structure of *PPD* at runtime based on the available compute and memory resources, facilitating its efficient deployment on various hardware platforms.

- An open-source implementation of *PPD*, accompanied by comprehensive evaluations on various models and benchmarks. Our experiments demonstrate that *PPD* achieves significant speed improvements with negligible memory overhead and reduced training cost.

## 2 BACKGROUND AND RELATED WORK

To enhance the inference speed of LLM, various approaches adopt an iterative **guess-and-verify** strategy to enable multi-token generation. In the guessing phase, potential future tokens are proposed at a faster speed than in traditional autoregressive implementations. Subsequently, a parallelized verification process assesses which guessed tokens should be accepted. Depending on how tokens are generated during the guess stage, these approaches can generally be categorized as *i)* speculative decoding and *ii)* parallel decoding.

### 2.1 SPECULATIVE DECODING

The guessing phase of speculative decoding adopts a lightweight draft model to generate multiple tokens at an increased speed (Kim et al., 2024). During the verification stage, the original LLM subsequently determines the acceptance of the guessed tokens. It is worth noting that both draft and original models still follow the auto-regressive inference scheme. The speedup comes from two factors: *i)* the draft model runs much faster than the original model and more tokens can be generated within the same time unit; and *ii)* token verification is executed concurrently, either by batching or by incorporating multiple candidates into a single input using customized sparse attention masks (Miao et al., 2024). Therefore, the overall speedup depends on the acceptance rate and the inference latency of draft models.

Building on the speculative decoding scheme, various studies have been conducted to further optimize its inference speed. To improve the accuracy of the draft model, *Eagle* (Li et al., 2024a) incorporates the hidden features into the draft model's forward pass. Recently, *Eagle-2* (Li et al., 2024b) enhances their approach using a context-aware dynamic tree construction. However, both *Eagle* and *Eagle-2* utilize a separate draft model for multi-token generation, diverging fundamentally from our prompt decoding approach. Moreover, their dynamic tree construction scheme is an orthogonal technique to our two-stage tree pruning method. *SpecInfer* (Miao et al., 2024) adopts a tree-based speculative inference and verification scheme, improving the diversity of speculation candidates. *Sequoia* (Chen et al., 2024) optimizes the sparse tree structure of speculative decoding by considering the capability of the underlying hardware platforms. Our tree pruning algorithm differs from *Sequoia* by accounting for two types of tokens in the tree: prompt tokens and guess tokens, whereas *Sequoia* only considers guess tokens. Furthermore, their methods require the storage and maintenance of a separate draft model, and there is extra complexity in designing an efficient draft model.

### 2.2 PARALLEL DECODING

To overcome the inherent limitations of autoregressive inference and the memory overhead associated with using a separate draft model, several attempts have been made to integrate both guessing and verification using one unified model. *Medusa*[1] (Cai et al., 2024) introduces language model (LM) heads at the final layer of the original LLM, facilitating the generation of multiple tokens in a single

---

[1]We categorize *Medusa* as parallel decoding because it only adopts LM heads instead of separate models.

forward pass. It also utilizes tree attention masks in its verification process to increase speed even further. To enhance token drafting with retrieval-augmented generation (Karpukhin et al., 2020), *Rest* (He et al., 2023) introduce retrieval-based decoding tailored for specific scenarios. Inspired by Jacobi decoding (Santilli et al., 2023) that adopts multiple special tokens to accelerate machine translation, *Lookahead Decoding* (Fu et al., 2024) improves upon this method by generating parallel n-grams and employing a caching memory pool. To capture more information while using multiple special tokens at distinct positions, *PaSS* (Monea et al., 2023) trains additional tokens with embedding layers for parallel decoding. Hierarchical parallel decoding (Liu et al., 2024) introduces the use of $[Fork]$ and $[Join]$ tokens, enabling parallel execution of multiple structural subroutines.

Our approach can be categorized as parallel decoding, with two novel features to distinguish it from other approaches: *1) PPD* trains the embeddings of parameterized ensemble prompt tokens, *2)* it utilizes a hardware-aware two-stage tree pruning algorithm for designing a sparse tree tailored to each hardware platform.

## 3 PARALLEL PROMPT DECODING (*PPD*)

The primary advantage of *PPD* lies in training embeddings for prompt tokens rather than developing a separate model. Our method integrates three substeps into a single decoding step, following the **guess-and-verify** strategy: (1) **candidate generation**, where multiple candidate continuations[2] are predicted by strategically inserting the prompt tokens into the input sequence. We adopt tree attention (Miao et al., 2024) to merge the processing of multiple candidates into a single forward pass; (2) **candidate verification**, where two verification schemes, exact matching (Fu et al., 2024) and typical acceptance (Cai et al., 2024), are implemented; (3) **candidate acceptance**, where validated candidates are integrated into the input and KV cache is updated accordingly. Figure 2 presents the inference scheme of combining generation and verification steps in a single forward pass.

### 3.1 PROMPT TOKENS

The prompt tokens are the key component of *PPD* to realize multi-token generation. Initially introduced by Lester et al. (2021) to adapt LLMs for specific tasks, prompt tokens are typically prepended to the input, with outputs generated in an autoregressive manner. In this work, we propose a novel approach of utilizing prompt tokens by strategically positioning them at locations where tokens are anticipated to be generated in parallel.

In the standard decoding process, the probability of predicting the next token is expressed as the conditional probability $p(y_{i+1}|x, y_{1:i})$, where $x$ is the input prompt, $y_{1:i}$ are the $i$ tokens generated so far, and $y_{i+1}$ is the next token to be predicted. For conventional parallel decoding techniques (Stern et al., 2018; Cai et al., 2024) that presume complete conditional independence among tokens decoded in a single step, the exact conditional probability is approximated by

$$p(y_{i+k+1}|x, y_{1:i+k}) = p_\theta(y_{i+k+1}|x, y_{1:i})$$

where $k > 0$ indicates the token distance.[3] However, we observe that as $k$ increases, the gap between the actual probability and its approximation expands, primarily due to the absence of relevant conditional dependencies. We argue that prompt tokens can bridge this gap by more accurately modeling the conditional probability as

$$p(y_{i+k+1}|x, y_{1:i+k}) = p_\theta(y_{i+k+1}|x, y_{1:i}, t_{i+1:i+k})$$

where $t_i$ is the prompt token with token distance $i$. Through this forward pass in the decoder layers, these causally linked prompt tokens facilitate the flow of information along the sequence of speculative tokens, thus restoring the conditional probability. We demonstrate the effectiveness of this approach in Section 5.2.

---

[2]A candidate token, also referred to as a "guess token", is a draft token generated from a prompt token.
[3]The token distance is the number of tokens between the last accepted token and the predicted token.

## 3.2 ENSEMBLE PROMPT TOKENS

Inspired by prompt ensembling (Lester et al., 2021), which uses multiple prompts to generate diverse responses and aggregates these to derive a single answer, we introduce the concept of ensemble prompt token (EPT). This additional abstraction allows us to decouple each prompt token from the fixed embedding dimension. For every prompt token, there exist multiple corresponding EPTs, each with its distinct embedding. We modify the attention mask to ensure that each $n^{\text{th}}$ EPT only depends on the corresponding $n^{\text{th}}$ EPTs from preceding prompt tokens. This selective visibility is maintained for both training and inference, where the guess token for each prompt token is determined by averaging the logits of its EPTs. The use of EPTs not only enables direct and flexible control over the trainable parameters, but also leads to an increase in prediction accuracy. The probability is approximated as $\frac{1}{n}\sum_{j=1}^{n} p_\theta(y_{i+k+1}|x, y_{1:i}, v^j_{i+1:i+k})$, where $v^j_{i+m}$ denotes the $j^{\text{th}}$ EPT at a token distance of $m$. Further details about EPTs can be found in Appendix D.

## 3.3 TRAINING

During training, only the embeddings of prompt tokens are changed, with the parameters of the original LLM remaining frozen. We adopt the following two training techniques:

**Random Insertion of Prompt Tokens:** Randomly inserting prompt tokens throughout the input sequence reduces contextual bias from appending them only at the end. This approach broadens the predictive capacity of prompt tokens beyond a limited vocabulary such as `<eos>` and punctuation.

**Knowledge Distillation:** To align the predictive behavior of prompt tokens with the original LLM, we employ knowledge distillation. Instead of using hard labels, prompt tokens are trained against the logits produced by the original LLM. Following Medusa (Cai et al., 2024), the loss function is formulated as:

$$L_{PD} = \frac{1}{N}\sum_{i=1}^{N} D_{KL}(P_i \parallel Q_i) \cdot \alpha^{i-1}, \tag{1}$$

where $D_{KL}$ is the KL divergence, $P_i$ is the predicted distribution of the $i^{\text{th}}$ prompt token, $Q_i$ is the corresponding distribution from the original LLM, and $\alpha$ is the decay ratio.

# 4 SPARSE TREE PRUNING

## 4.1 CUSTOMIZED SPARSE TREE ATTENTION

Tree attention, introduced by SpecInfer (Miao et al., 2024), increases the expected acceptance rate by considering the top-k candidates from a single decoding step. In their approach, the input is structured as a tree, where each level of the tree corresponds to a specific output position. An attention mask is applied to the tree-structured input, allowing the model to process multiple candidates efficiently without increasing the batch size.

To improve the efficiency and performance of LLM inference, this paper proposes a novel sparse tree customized for *PPD*, which prioritizes candidates in the tree structure with higher prediction accuracy. A key difference from previous works (Cai et al., 2024; Chen et al., 2024) is the appending of a sequence of prompt tokens to each guess token. The length of the prompt token sequence decides the maximum depth of the speculative tree at the next decoding step. To further hide the latency introduced by the extra prompt tokens, we propose a novel tree pruning algorithm (Section 4.2) that optimizes the number of prompt tokens at each guess token.

## 4.2 TWO-STAGE TREE PRUNING ALGORITHM

As depicted in Figure 3, our tree pruning algorithm consists of two stages: an offline static tree pruning phase and an online hardware-aware tree optimization phase. These two stages are applied subsequently to reduce the amount of computation involved in *PPD* multi-token generation.

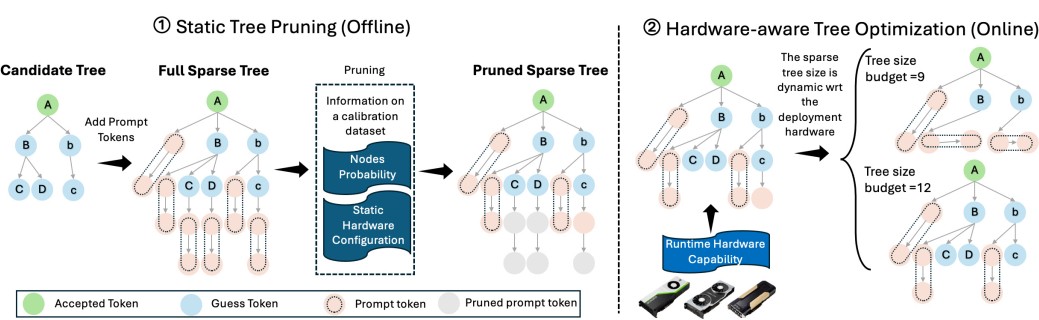

Figure 3: Illustration of Tree Pruning Pipeline. The tree structure is optimized as a result of pruning.

### 4.2.1 STATIC TREE PRUNING

The first stage, static tree pruning, is applied offline prior to runtime deployment. The goal is to reduce the number of prompt tokens in the tree to achieve the desired tree size. As shown on the left side of Figure 3, the tree pruning process consists of three key steps:

1. **Candidate Trees Construction:** Building trees using only candidate tokens at varying depths, employing the algorithm from Medusa (Cai et al., 2024) and Sequoia (Chen et al., 2024) to maximize $f(T_k)$.

2. **Prompt Tokens Appending :** Attaching the maximum allowable prompt tokens to each candidate token from the first step.

3. **Greedy Prompt Token Removal:** Removing a prompt token greedily to maximize expected amortized acceptance lengths, continuing until the desired prompt token budget is reached.

Each guess token in the tree is appended with a sequence of prompt tokens, with each prompt token corresponding to a unique output position. The length of this sequence determines the tree's maximum depth at the next decoding step. Thus, removing a prompt token at a guess token reduces the maximum tree depth at the next decoding step if this guess token is accepted in the current step. Let $p_c$ represent the acceptance probability of guess token $c$, and $f_d$ denote the expected acceptance length with $d$ prompt tokens before removal. The decrease in expected acceptance length, $\Delta F$, due to removing a prompt token at $c$ is given by $\Delta F = p_c \cdot (f_d - f_{d-1})$. More details are discussed in Appendix A.

### 4.2.2 HARDWARE-AWARENESS TREE OPTIMIZATION

Given that hardware platforms differ in terms of memory, computational resources, and runtime capabilities, we propose a hardware-aware tree optimization to maximize the overall performance of *PPD*. As shown on the right of Figure 3, this optimization adjusts the tree size budget based on the performance characteristics of the target hardware.

To achieve this, we define two key functions:

1. Acceptance length $\tau(n)$ (hardware-independent) and
2. Forward pass latency $L_{fp}(n)$ (hardware-dependent).

The speedup ratio, $\text{Speedup}(n) = \frac{\tau(n)}{L_{fp}(n)}$, is estimated using a validation dataset, with $\tau(n)$ evaluated once and $L_{fp}(n)$ tested on different hardware platforms. We then choose the tree size budget that maximizes $\text{Speedup}(n)$ based on the measured runtime latency on the specific hardware platform. To eliminate runtime overhead, hardware latency profiling is conducted during idle periods.

## 5 EXPERIMENTS

**Models and testbeds**. We conducted all the experiments using MobileLLaMA-1.4B (Chu et al., 2023), Vicuna-7B and Vicuna-13B (Chiang et al., 2023). We used 3 prompt tokens and 1 EPT per

prompt token for all inference experiments. The inference throughputs of the models are evaluated on a single NVIDIA A100 GPU with 40GB of memory and a GeForce RTX 4090 using a batch size of 1 and FP16 precision. Further details about the experimental setup can be found in Appendix F.

**Training**. We froze all trainable parameters of the original LLM. Prompt token embeddings were trained using distillation logits generated from the ShareGPT dataset (ShareGPT, 2023), with a maximum context length of 1024, a cosine learning rate scheduler starting at 0.01, and no warmup. Prompt token embeddings are initialized with normal text token embeddings. For each model, the same set of prompt tokens is used across experiments to demonstrate its generalizability.

**Datasets**. We assess the throughput performance of *PPD* across various tasks and datasets. Specifically, we evaluated *PPD* using the MT-Bench dataset (Zheng et al., 2023), which contains multi-turn questions with a range of topics, in both non-greedy (temperature follows the default configuration) and greedy settings (temperature=0). We used the GSM8K (Cobbe et al., 2021) and HumanEval (Chen et al., 2021) datasets only in the greedy setting. The GSM8K dataset consists of grade school math problems and we used the first 500 questions of the test split for our evaluations. HumanEval includes coding tasks, for which we set a maximum new token limit of 512 to control the length of the generated sequences. We used the Alpaca (Li et al., 2023) dataset as the validation dataset to produce the latencies and acceptance lengths used for sparse tree pruning.

## 5.1 Speedup Comparison with Parallel Decoding Methods

We compare the speedup ratios of *PPD* with state-of-the-art parallel decoding methods on MT-Bench in non-greedy settings in Figure 4. *PPD* achieves speedups up to 13.8% higher than Medusa and between 2 times and 3 times higher than other parallel decoding methods. We examine the factors contributing to the enhanced speedup ratios and other performance metrics, as presented in Table 1. The reasons for the increase in speedup ratios are twofold. Firstly, *PPD* produces candidate tokens with a higher acceptance rate than Medusa when utilizing a sparse tree of the same size. Notably, *PPD* continues to achieve a comparable or slightly better acceptance rate even when employing a much smaller sparse tree – ranging from one-third to half the size. Secondly, *PPD* benefits from lower forward pass latency due to its ability to use smaller sparse tree sizes and hence shorter input lengths. *PPD* also eliminates the computational overhead associated with separate decoding heads. *PPD* maintains the same output quality, achieving about the same score on MT-Bench while using significantly fewer trainable parameters.

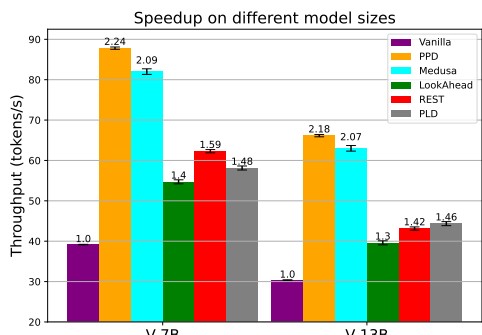

Figure 4: Comparative evaluation of latency speedup between *PPD* and other parallel decoding methods. The experiments were conducted using the MT-Bench dataset, with the temperature set to MT-Bench's default configuration for Medusa and *PPD*.

Figure 5 displays the throughput of *PPD* on MT-Bench, HumanEval, and GSM8K with temperature equal to 0. *PPD* achieves consistent walltime speedup ratios from 2.12× to 2.49× on different GPUs, which demonstrates that prompt tokens generalize well on different tasks. In general, *PPD* performs better in coding and math reasoning tasks, achieving speedups between 2.21× and 2.49×. This can be attributed to the fact that both code and math equations often contain fixed patterns and repetitive symbols, which narrows the range of plausible candidates and simplifies the prediction. We also found that with typical acceptance, the speedup increases with temperature. Another notable trend is that smaller models, such as Vicuna-7B, generally achieve more significant speedup ratios as compared to larger models, like Vicuna-13B. *PPD* aims to generate more tokens per step, which comes with increased computational demands. For larger models that already require substantial computational resources, it is necessary to limit the size of the sparse tree to avoid exceeding the GPU's utilization cap and causing increased latency. As a result, the number of tokens accepted per step is reduced, leading to lower speedups. However, this can be amortized when using more powerful GPUs than the NVIDIA A100 and the RTX 4090, such as NVIDIA H100.

| Model | Method | $T$ | $\tau$ | $L_{\text{fp}}$ (s) | Quality | $P_{\text{tr}}$ (%) | $S_{\text{tr}}$ | $S_{\text{input}}$ |
|-------|--------|-----|--------|-----|---------|------|------|--------|
| **M** | Vanilla | 50.2 | 1.00 | **0.020** | - | NA | NA | 1 |
|       | *PPD* | **108.7** | **2.43** | 0.022 | *Same* | **4.50**$e^{-4}$ | (10,84,89) | (40,285,285) |
| V-7B | Vanilla | 39.2 | 1.00 | **0.026** | **5.99** | NA | NA | 1 |
|      | Medusa | 82.0 | 2.51 | 0.0307 | 5.98 | 8.07 | 63 | 63 |
|      | *PPD* | **88.0** | **2.54** | 0.029 | 5.93 | **1.82**$e^{-4}$ | (10,33,34) | (40,105,105) |
| V-13B | Vanilla | 30.4 | 1.00 | **0.0330** | **6.38** | NA | NA | 1 |
|       | Medusa | 63.4 | **2.59** | 0.0408 | - | 5.52 | 63 | 63 |
|       | *PPD* | **66.1** | 2.44 | 0.0379 | 6.32 | **7.87**$e^{-5}$ | (10,20,20) | (40,60,60) |

Table 1: Comparative performance metrics of MobileLLaMA (M) for greedy setting, Vicuna-7B (V-7B) and Vicuna-13B (V-13B) for non-greedy setting using different decoding methods. The table details throughput ($T$ in tokens/s), average accept lengths ($\tau$ in tokens), forward pass latency ($L_{\text{fp}}$ in seconds), quality scores on MT-benchmark, percentages of additional trainable parameters ($P_{\text{tr}}$) and input lengths ($S_{\text{input}}$) after the prefilling phase. The sparse tree size ($S_{\text{tr}}$) of *PPD* varies at different time steps as a consequence of different numbers of prompt tokens at each guess token, hence represented as tuples. *Same* means the output matches with that of the original LLM.

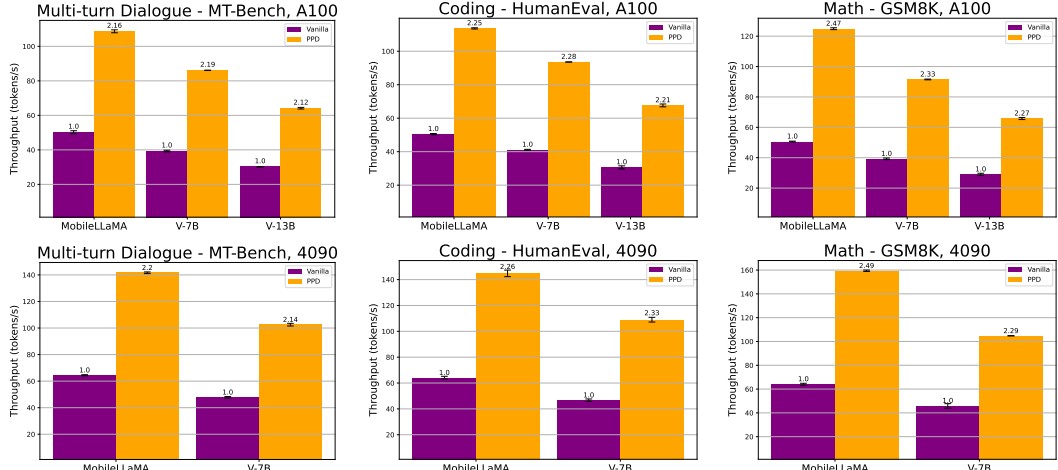

Figure 5: Throughput of *PPD* and vanilla models across different tasks (**multi-turn dialogue, coding, and math**). The temperature for experiments is set to 0 and the generated output of *PPD* exactly matches that of the original LLM. We do not show the results of Vicuna-13B on RTX 4090 as it does not fit into the GPU memory.

## 5.2 LONG-RANGE TOKEN PREDICTION

For a specific sparse tree, the accumulative accuracy provides a theoretical upper bound for the number of generated tokens per step and the maximum possible speedup ratio. Hence, maximizing

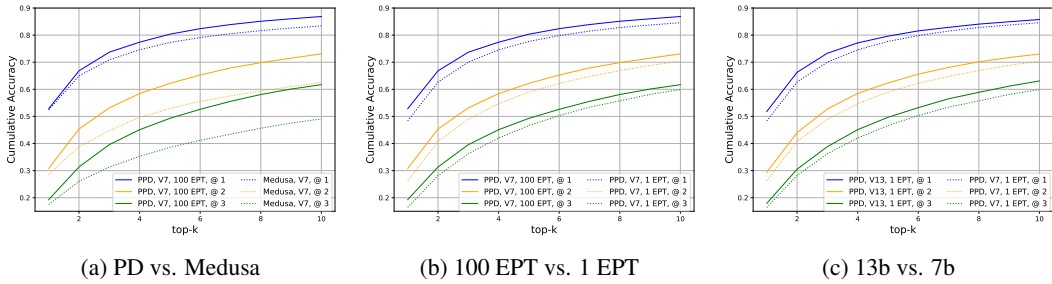

(a) PD vs. Medusa      (b) 100 EPT vs. 1 EPT      (c) 13b vs. 7b

Figure 6: Accumulative accuracy comparisons across different model configurations and prediction distances. 'V7' for Vicuna-7B, and 'V13' for Vicuna-13B. The notation '@$i$' refers to a token distance of $i$. '100 EPT' represents 100 EPTs per prompt token. Accumulative accuracy is defined as top-k accuracy (*e.g.*, a prediction is correct if the top-k candidates contain the ground truth). These measurements were obtained from the Alpaca Eval dataset with a maximum of 20 steps.

accumulative accuracy is crucial for the effectiveness of *PPD*. Figure 6 demonstrates the accumulative accuracy of the tokens predicted at various positions. We summarize the following three key insights from the results.

***PPD* excels at predicting more distant tokens.** As depicted in Figure 6a, *PPD* consistently outperforms Medusa in accuracy across all token positions. The accuracy gap between *PPD* and Medusa widens with the increased token distance (*e.g.*, the top-10 accuracy difference is 0.03 for the 'next next' word versus 0.12 for the 'next next next' word). This improvement can be attributed to *PPD*'s ability to partially recover conditional dependency information through causally connected prompt tokens.

***PPD* performs well at generating a broader array of plausible token candidates.** For example, in predicting the token at a token distance of 3, the top-10 candidates exhibit an accuracy improvement of 0.1 over Medusa, compared to only 0.02 for the top-1 candidate. This demonstrates the value of using tree attention and the largest viable tree size during inference, as multiple candidate continuations further boost accuracy improvement.

**Multiple EPTs per prompt token and larger model sizes yield modest improvements in prediction accuracy**. Figure 6b shows that using 100 EPTs per prompt token leads to accuracy improvement, ranging from 0.018 to 0.045. Figure 6c displays that *PPD* with Vicuna-13B outperforms Vicuna-7B with an accuracy gain of 0.011∼0.038. This increase is due to Vicuna-13B's greater embedding dimensions and deeper layers, which enhance the expressive power of prompt tokens. However, these gains are modest and can be offset by the increased computational burden of larger models.

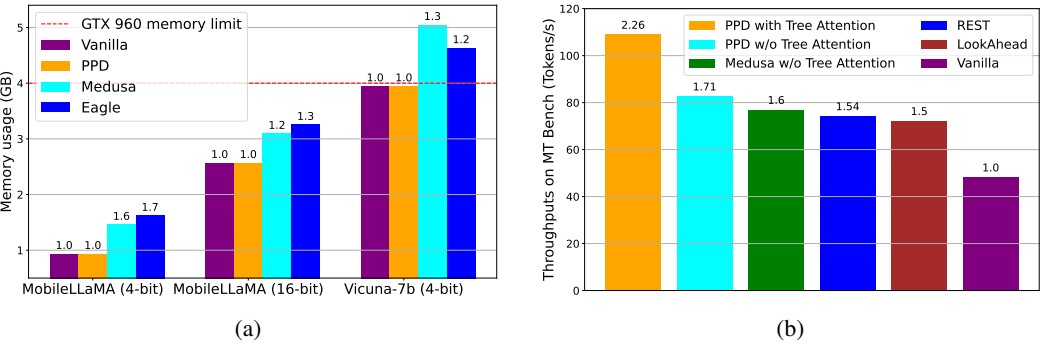

|        |        |
|:------:|:------:|
| (a)    | (b)    |

Figure 7: (a) Memory usage of PPD and other baseline methods including Vanilla, Medusa, and Eagle; (b) Throughput comparison of *PPD* with other parallel decoding approaches. We control the use of tree attention in some approaches for ablation analysis.

## 5.3 MEMORY AND TRAINING EFFICIENCY

**Memory efficiency.** As shown in Figure 7a, we compare the memory overhead of *PPD* with the leading parallel decoding (Medusa) and speculative decoding approaches (Eagle). The memory overhead of PPD is just 0.004% of Medusa's and 0.007% of Eagle's. This efficiency stems from the efficient use of embeddings in *PPD*, which are significantly smaller than decoding heads and draft models, both of which scale with vocabulary size.

**Training efficiency.** Table 2 compares the training times of *PPD* with parallel and speculative decoding methods. *PPD* is trained until its evaluation accuracy of top-10 candidates surpasses that of Medusa on Alpaca Eval. Notably, *PPD* surpasses Medusa in evaluation accuracy while training in less than half the time, demonstrating its great potential to reduce training cost.

| Method | Training Time |
|:------:|:------:|
| *PPD* (Ours) | **0.52 hours** |
| Medusa | 1.24 hours |
| Eagle | 1-2 days |

Table 2: Training time of *PPD*, Medusa, and Eagle, on 4 A100 GPUs. *PPD* takes less than half of the time compared to Medusa.

### 5.4 ABLATION STUDY

**Tree Attention.** As illustrated in Figure 7b, tree attention boosts the speedup ratio of *PPD* by an additional 32%, indicating that *PPD* generates accurate top-k predictions. Even without the use of tree attention, *PPD* still outperforms all other parallel decoding methods, achieving up to a 14% higher speedup ratio, demonstrating the effectiveness of our approach.

**Sparse Tree Pruning Algorithm.** Figure 8a shows that the pruned sparse trees consistently achieve longer acceptance lengths compared to naive and random ones across varying sizes. The acceptance length for pruned sparse trees shows a steady increase as the tree size extends, suggesting its good scalability. The convergence of pruned and naive sparse trees at larger sizes suggests a structural similarity emerging from constraints in tree depth and tree node count.

**Hardware-aware Tree Size.** Figure 8b presents the theoretical speedup across different GPUs. Figure 8c validates that the optimal sparse tree size, derived from theoretical speedup models, indeed results in the greatest actual speedup observed.

***PPD* + Speculative Decoding.** As an orthogonal optimization in accelerating LLMs, *PPD* can be easily integrated with speculative decoding (Kim et al., 2024). To demonstrate this, we applied *PPD* to Vicuna-68M (Yang et al., 2024) and used it as the draft model for Vicuna-7B. This combination resulted in a speedup of up to 1.22× for speculative decoding on Vicuna-7B compared to using speculative decoding alone.

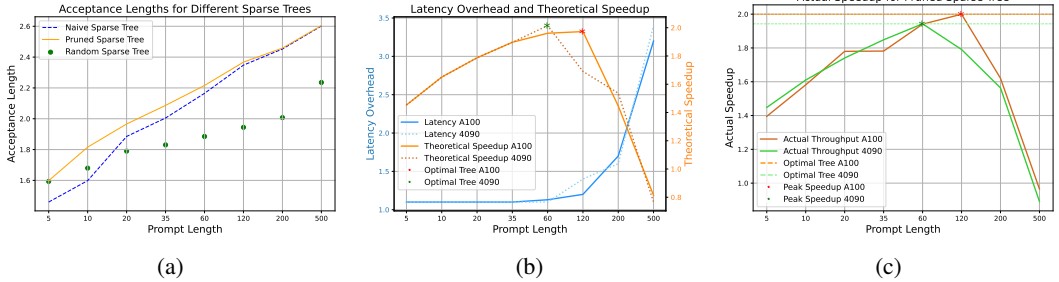

(a)                                          (b)                                          (c)

Figure 8: Evaluation of Sparse Tree Pruning Algorithm. The naive sparse tree in (a) applies the same number of prompt tokens to each guess token, while the pruned sparse tree follows our pruning algorithm. The random sparse tree allocates prompt token budget randomly. The theoretical speedup in (b) is calculated as the ratio of acceptance lengths (hardware-independent) to latency overhead (hardware-dependent). The optimal tree size is obtained from the peak value of the theoretical speedup. The latencies in (b) are obtained from inference on the same prompt for 512 forward passes. (c) shows the actual speedup obtained by running inference on different GPUs with different tree lengths on Alpaca Eval dataset.

## 6 CONCLUSION

We introduced *PPD*, a memory-efficient, cost-effective, and powerful parallel decoding method that incorporates a hardware-aware online tree optimization. Utilizing specially trained prompt tokens to predict long-range tokens accurately, *PPD* achieves a speedup of up to 2.49× in inference while employing only 0.0002% additional trainable parameters without incorporating new models or architectural components. We showcased that *PPD* offers a novel perspective on the capabilities of parallel decoding. Importantly, it could be synergized with other speculative or parallel decoding techniques to expedite inference even further. We hope that by open-sourcing the code base (upon acceptance of the paper), *PPD* can help the community further advance the performance of real-world deployment of the current and future decoder-based LLM models.

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

# Supplementary Material
## Hardware-Aware Parallel Prompt Decoding for Memory-Efficient Acceleration of LLM Inference

## Table of Contents

# A   DETAILED TREE CONSTRUCTION ALGORITHM

We follow the same optimal sparse tree construction approach as in Medusa (Cai et al., 2024).

**Definition A.1.** Let $m$ be the maximal number of prompt tokens per tree node. The sparse tree $T$ can exist in $m$ states, each represented by $T_k$ corresponding to state $s_k$, where $1 \leq k \leq m$. Let $\mathrm{C}(T_k)$ denote the subtree of $T_k$ composed solely of candidate tokens. The maximum depth of $\mathrm{C}(T_k)$ is $k$.

**Proposition A.1.** For a sparse tree state $T_k$, where each candidate token $v$ follows a path $\mathrm{Path}(v)$ from the root, and the acceptance probability $p_k$ at each path position $k$, the expected number of tokens $f(T_k)$ generated is given by $f(T_k) = \sum_{v \in \mathrm{C}(T_k)} \prod_{i \in \mathrm{Path}(v)} p_i$, where $\prod_{i \in \mathrm{Path}(v)} p_i$ represents the contribution of a token $v$ to the expected number of tokens.

We then propose an approximation of the amortized number of tokens generated, by considering the tokens generated at the current and the next decoding step.

**Proposition A.2.** The expected total number of tokens $F(T_k)$ generated for the sparse tree state $F(T_k)$ at the current and the next decoding step is given by $F(T_k) = f(T_k) + \sum_{i=1}^{m} p(s_i|s_k)f(T_i)$, where $p(s_i|s_k)$ represents the state transition probability from state $s_k$ to state $s_i$.

We are now ready to introduce Proposition A.3, which we use in the pruning algorithm.

**Proposition A.3.** For a sparse tree state $T_k$ with candidate subtree $c_k = \mathrm{C}(T_k)$, the change in expected total tokens $F(T_k)$ due to the removal of a prompt token at candidate token $c$ is given by $\Delta F = p(c) \cdot (f(T_i) - f(T_{i-1}))$, where $p(c)$ is the acceptance probability of candidate $c$, $i$ denotes the number of prompt tokens prior to removal. We assume that $i > 1$.

We now introduce the formulation of the real amortized number of tokens generated.

**Proposition A.4.** The amortized number of tokens $R(T_k)$ generated for the sparse tree state $F(T_k)$ is given by $R(T) = \sum_{i=1}^{m} p(s_i)f(T_i)$, where $p(s_i)$ is the steady-state probability of state $s_i$, and $f$ is the function defined in Proposition A.1.

The sparse tree construction algorithm can now be formulated as finding the sparse tree $T$ with $n_c$ candidate tokens and $n_p$ prompt tokens to maximize $R(T)$:

$$c(n_c, n_p) = \max_{T, |C(T)|=n_c, |T|=n_c+n_p} R(T).$$

For a fixed tree size $n$, we explore all combinations of $n_c$ and $n_p$ where $n = n_c + n_p$, to identify the sparse tree that maximizes $R(T_k)$.

# B   TRAINING LOSS

We study the training loss of *PPD* with different EPTs. Figure 9a shows that, with 3 prompt tokens and 1 EPT, the initial loss is quite high, starting above 5. There is a sharp decrease in loss within the first epoch, dropping below 2. After this initial drop, the loss stabilizes and oscillates around a value slightly below 2 for the remainder of the training epochs (up to epoch 12). The loss oscillations remain within a narrow range, indicating consistent performance. The fluctuation can be attributed to the insertion of prompt tokens at random positions. On the other hand, Figure 9b, with 3 prompt tokens and 100 EPTs, shows the initial loss starting below 3, significantly lower than *PPD* with 1 EPT. Similarly, there is a sharp decrease within the first epoch, with the loss dropping to around 2.5. However, unlike *PPD* with 1 EPT, the loss continues to decrease gradually over the epochs, showing a downward trend. This suggests that increasing the number of EPTs improves the model's learning capacity and reduce training loss more effectively over time.

# C   GENERALIZABILITY OF PROMPT TOKENS TO DIFFERENT TASKS

While original prompt tuning tailors LLMs for specific downstream tasks, our prompt tokens are task-agnostic. To demonstrate their generalizability, Figure 10 shows the prediction accuracy of a single set of prompt tokens across three different datasets. Trained on the ShareGPT dataset, these tokens generalize effectively to unseen tasks. We also report the prediction accuracy for using 5 prompt tokens.

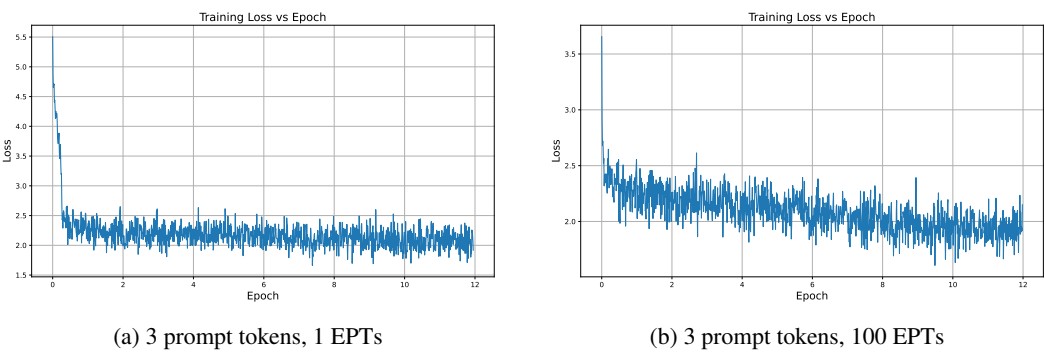

(a) 3 prompt tokens, 1 EPTs        (b) 3 prompt tokens, 100 EPTs

Figure 9: Training Loss

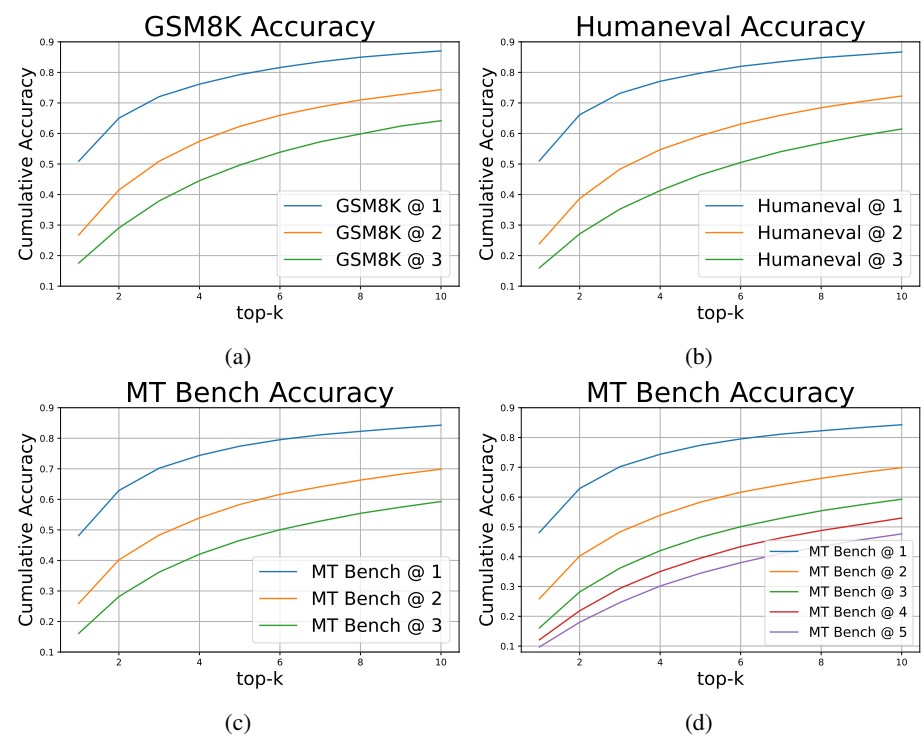

Figure 10: Evaluation accuracy of the same set of prompt tokens on (a) GSM8K dataset, (b) HumanEval dataset, (c) MT-Bench dataset, and (d) prediction accuracy of 5 prompt tokens.

## D EXTENDED ABLATION STUDY

### D.1 EFFECT OF EPTS ON PREDICTION ACCURACY

Table 3 presents the prediction accuracy of *PPD* using different EPTs. The results indicate that increasing the number of EPTs generally enhances the prediction accuracy of PPD, particularly for long-range token predictions. Higher EPT numbers (e.g., 100 and 50) consistently produce better prediction accuracy compared to lower EPT numbers.

### D.2 IMPACT OF KNOWLEDGE DISTILLATION (KD), EPOCHS, AND BATCH SIZE ON PREDICTION ACCURACY

Table 4 summarizes our results with different settings. We analyze the effect of each factor on the prediction accuracy in the following discussion.

| EPT | @1 Top-1 | @1 Top-5 | @2 Top-1 | @2 Top-5 |
|-----|----------|----------|----------|----------|
| 100 | 0.506 | 0.794 | 0.276 | 0.602 |
| 50  | 0.502 | 0.791 | 0.281 | 0.604 |
| 20  | 0.501 | 0.791 | 0.276 | 0.607 |
| 10  | 0.494 | 0.786 | 0.273 | 0.600 |
| 5   | 0.499 | 0.787 | 0.265 | 0.596 |
| 2   | 0.486 | 0.777 | 0.259 | 0.583 |
| 1   | 0.472 | 0.771 | 0.248 | 0.576 |

Table 3: Prediction Accuracy of *PPD* with different EPTs. '@i' denotes a token distance of i. 'Top-k' denotes the top-k prediction accuracy. The results are obtained on Alpaca dataset with 20 steps.

| EPT | KD | Epoch | Batch | @1 Top-1 | @1 Top-5 | @2 Top-1 | @2 Top-5 |
|-----|-----|-------|-------|----------|----------|----------|----------|
| 100 | Yes | 1  | 4 | 0.504 | 0.793 | 0.273 | 0.598 |
| 100 | Yes | 2  | 4 | 0.512 | 0.797 | 0.288 | 0.611 |
| 100 | Yes | 6  | 4 | 0.520 | 0.802 | 0.302 | 0.620 |
| 100 | Yes | 8  | 4 | 0.524 | 0.804 | 0.307 | 0.619 |
| 100 | Yes | 10 | 4 | 0.523 | 0.804 | 0.305 | 0.623 |
| 100 | Yes | 12 | 4 | 0.525 | 0.805 | 0.308 | 0.625 |
| 100 | No  | 12 | 4 | 0.506 | 0.794 | 0.276 | 0.602 |
| 100 | Yes | 12 | 1 | 0.530 | 0.809 | 0.309 | 0.626 |
| 1   | Yes | 12 | 1 | 0.484 | 0.775 | 0.259 | 0.581 |
| 1   | Yes | 2  | 4 | 0.474 | 0.773 | 0.247 | 0.574 |
| 1   | Yes | 6  | 4 | 0.480 | 0.773 | 0.250 | 0.580 |
| 1   | Yes | 8  | 4 | 0.484 | 0.778 | 0.257 | 0.583 |
| 1   | Yes | 10 | 4 | 0.482 | 0.777 | 0.257 | 0.584 |
| 1   | Yes | 12 | 4 | 0.485 | 0.779 | 0.261 | 0.586 |
| 1   | No  | 12 | 4 | 0.472 | 0.771 | 0.248 | 0.576 |

Table 4: Prediction Accuracy for *PPD* with and without knowledge distillation (KD) for different EPTs, epochs, and batch sizes.

### D.2.1 TRAINING EPOCHS

We first investigate the effect of the number of training epochs on prediction accuracy. For models using 100 EPTs with KD enabled and a batch size of 4, we observe a steady improvement in prediction accuracy as the number of epochs increases. Specifically, the Top-1 accuracy at a 1-token distance increases from 0.504 at 1 epoch to 0.525 at 12 epochs, while the Top-5 accuracy at a 1-token distance improves from 0.793 to 0.805. Similarly, Top-1 accuracy at a 2-token distance increases from 0.273 to 0.308, and Top-5 accuracy at a 2-token distance improves from 0.598 to 0.625 over the same range of epochs. This trend demonstrates the positive impact of prolonged training on the performance of *PPD* when KD is applied.

### D.2.2 KNOWLEDGE DISTILLATION

When KD is not applied, as shown for 100 EPTs at 12 epochs with a batch size of 4, the performance metrics are generally lower. The improvement in prediction accuracy with KD is up to 12%. This suggests that KD contributes significantly to prediction accuracy for *PPD*.

### D.2.3 EFFECT OF BATCH SIZE

We also examine the impact of batch size on the prediction accuracy. For the model trained with 100 EPTs, KD enabled, and 12 epochs, reducing the batch size from 4 to 1 results in a slight improvement in prediction accuracy up to 1%.

## D.3 PREFIX TUNING + PROMPT TOKEN

Prefix tuning (Li & Liang, 2021), similar to prompt tuning, provides a parameter-efficient approach to fine-tune a pre-trained model. Unlike prompt tuning, it modifies the KV cache of every attention layer by prepending trained vectors. We hypothesize that the combination of prefix tuning and prompt tokens can lead to greater learning capacity and higher prediction accuracy. This hypothesis is based on the intuition that prompt tokens should see a different context than the input tokens when predicting long-range tokens. For example, if the input sequence is "Once upon a time", then enhancing the input with a prompt template might provide more suitable semantic context for long-range prediction. An enhanced input like "Predict the next-next token. Once upon a time" might empower the prompt token to predict the correct next-next token. Prefix tuning serves as the prompt template to enhance the hidden states visible to the prompt tokens.

Figure 11: 'P1' is the prefix token for the prompt token 'S1' and 'P2' for 'S2'. 'C' is the input token. The green tick means visibility during attention calculation. For instance, 'S1' can see 'P1' but cannot see 'P2'. 'C' does not see any prefix tokens so the generated output corresponding to 'C' is not altered by the use of prefix tuning.

To retain the original model's distribution, we modify the attention mask so that prefix tokens are only visible to prompt tokens. This ensures that we can generate outputs that preserve the original model's distribution. We posit that prompt tokens at different positions should see different contexts so we allow a prompt token at a specific position to see a distinct set of prefix tokens, as shown in Figure 11.

| Prefix Tuning | @1 Top-1 | @1 Top-5 | @2 Top-1 | @2 Top-5 |
|---|---|---|---|---|
| No | 0.485 | 0.779 | 0.261 | 0.586 |
| Yes | 0.412 | 0.738 | 0.204 | 0.541 |

Table 5: Prediction Accuracy of *PPD* with and without prefix tuning. 1 EPT is used for all models and 1 prefix token is used for prefix tuning.

Table 5 compares the prediction accuracy of *PPD* with and without the use of prefix tuning. The results show that the models without prefix tuning outperform those with prefix tuning up to 28%, which suggests that, in this setup, prefix tuning does not enhance the prediction accuracy of PPD. Instead, it appears to degrade performance, potentially due to the complexity introduced by modifying the KV cache of attention layers with the prefix token. Unlike prompt tokens, prefix tokens do not interact with input tokens, meaning they do not change dynamically through the transformer layers based on the input context. This lack of interaction and dynamic adjustment could be a factor contributing to the decreased prediction accuracy observed with prefix tuning.

## D.4 CUSTOM DECODING HEADS + PROMPT TOKEN

It has been demonstrated that a fine-tuned decoding head alone can effectively predict long-range tokens (Stern et al., 2018; Cai et al., 2024). Thus, we hypothesize that combining a separately

fine-tuned decoding head with prompt tokens might further enhance the potential of *PPD*. As shown in Figure 12, we trained a separate decoding head to transform only the hidden states of prompt tokens into logits. A key distinction from Medusa is that this decoding head is responsible for generating tokens at multiple positions, rather than just one.

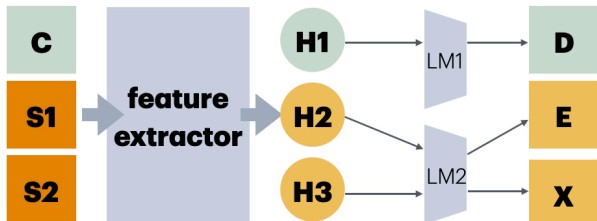

Figure 12: Custom decoding head with *PPD*. The feature extractor refers to the LLMs without the decoding heads. 'H1' is the generated hidden state for the input token 'C'. 'H2' is the hidden state for the prompt token 'S1' and 'H3' for 'S2'. 'LM1' is the original LLM's decoding head and it takes in the hidden states of input tokens. 'LM2' is the custom decoding heads for *PPD* and only takes in the hidden states of prompt tokens.

We propose two training methods. In the first method, the custom decoding head and prompt tokens are trained together from scratch in a single stage. In the second method, the prompt tokens are initially trained for 2 epochs, followed by training both the prompt tokens and the decoding head with a smaller learning rate in a two-stage process.

| Method Name | @1 Top-1 | @1 Top-5 | @2 Top-1 | @2 Top-5 |
|---|---|---|---|---|
| *PPD* without custom decoding head | 0.485 | 0.779 | 0.261 | 0.586 |
| *PPD* with custom decoding head (1-stage) | 0.385 | 0.614 | 0.229 | 0.482 |
| *PPD* with custom decoding head (2-stage) | 0.506 | 0.795 | 0.276 | 0.602 |

Table 6: Prediction Accuracy of *PPD* with and without custom decoding head. 1 EPT is used for all models. 1-stage and 2-stage refer to the training strategies of custom decoding head.

Table 6 presents the prediction accuracy of *PPD* with and without a custom decoding head. When trained using the single-stage method, *PPD* with the custom decoding head shows a 12%-21% decrease in prediction accuracy compared to the baseline *PPD* without the custom decoding head. This suggests that the single-stage approach does not result in stable or effective training.

In contrast, the two-stage training method results in a limited improvement of 2.1%-4.3% in prediction accuracy compared to the baseline. This suggests that adding a custom decoding head may not be necessary, given the additional trainable parameters and the limited improvement in prediction accuracy.

### D.5   ATTENTION MASKING FOR EPTS

In this paper, we proposed a specialized attention mask for EPTs to achieve the effect of prompt ensemble. However, there are alternative masking strategies available. Here, we describe and compare three types of attention masks that we implemented and experimented with.

### D.5.1   ENSEMBLE ATTENTION MASKING

The ensemble attention masking is the masking strategy we previously described. In this approach, EPTs are divided into $n$ disjoint groups, where $n$ is the number of EPTs per prompt token. All $k^{th}$ EPTs across prompt tokens are placed in the same group. An EPT $v$ in group $i$ can only attend to EPTs that meet the following two criteria: 1) they must belong to group $i$, and 2) their position indices must be smaller than the position index of $v$. Since this masking strategy effectively averages the results of disjoint groups of EPTs, we refer to it as the "ensemble attention masking". Figure 13a provides an example of the ensemble attention masking.

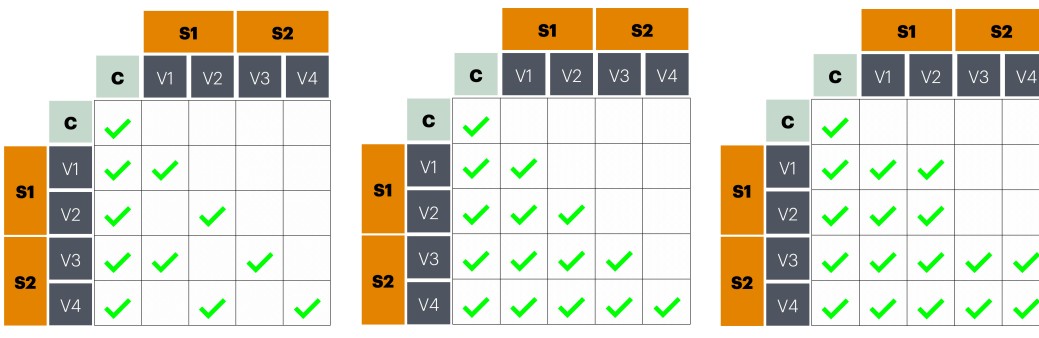

(a) Ensemble Attention Mask     (b) Decoder-like Attention Mask     (c) Encoder-like Attention Mask

Figure 13: Different Mask Strategies for EPTs. 'C' is an input token. 'V1' and 'V2' are the EPTs for prompt tokens 'S1' and 'V3' and 'V4' for 'S2'.

### D.5.2 DECODER-LIKE ATTENTION MASKING

Decoder-like attention masking is a simple strategy where EPTs can only attend to EPTs with smaller position indices. This results in a triangular-shaped attention mask, similar to the one used in decoder layers, hence the name "decoder-like attention masking". Figure 13b provides an example of this masking strategy.

### D.5.3 ENCODER-LIKE ATTENTION MASKING

In encoder-like attention masking, an EPT corresponding to a prompt token $P$ can attend to all EPTs with smaller position indices as well as all EPTs associated with $P$. This allows EPTs to see both preceding and succeeding EPTs, similar to the token visibility in an encoder layer, hence the name "encoder-like attention masking". Figure 13c illustrates this masking strategy.

### D.5.4 RESULTS

| Method Name | @1 Top-1 | @1 Top-5 | @2 Top-1 | @2 Top-5 |
|---|---|---|---|---|
| *PPD* with ensemble attention mask | 0.506 | 0.794 | 0.276 | 0.602 |
| *PPD* with decoder attention mask | 0.465 | 0.755 | 0.262 | 0.572 |
| *PPD* with encoder attention mask | 0.473 | 0.765 | 0.256 | 0.573 |

Table 7: Prediction Accuracy of *PPD* with different attention masking strategies for EPTs. 100 EPT is used for all models.

The results in Table 7 indicate that the ensemble attention mask outperforms the other masking strategies. In comparison, the *PPD* with decoder attention mask shows 4.9%-8.0% lower prediction accuracy. The *PPD* with encoder attention mask also underperforms in prediction accuracy relative to the ensemble attention mask by 3.7%-7.2%.

These results suggest that the ensemble attention mask is the most effective strategy among the three, likely due to its ability to effectively average the votes of disjoint groups of EPTs, thereby improving prediction accuracy. The decoder-like and encoder-like attention masks, while simpler, do not provide the same level of performance, indicating that the structure and specificity of the ensemble attention mask better facilitate accurate long-range token prediction. Additionally, ensemble attention masking is more sparse, which offers greater potential for optimization.

### D.6 AGGREGATION METHOD FOR EPTS

In addition to simply averaging the logits from EPTs, we explored more advanced aggregation methods. For instance, we applied learned weights to aggregate the logits. The final logit $p$ can be

expressed as:

$$p = \sum_{i=1}^{n} w_i \cdot p_i,$$

where $n$ is the number of EPTs and $w_i$ is the learned scalar weight for the $i^{th}$ EPT.

| Aggregation Method | @1 Top-1 | @1 Top-5 | @2 Top-1 | @2 Top-5 |
|---|---|---|---|---|
| Average | 0.506 | 0.794 | 0.276 | 0.602 |
| Learned Weight | 0.503 | 0.779 | 0.250 | 0.576 |

Table 8: Prediction Accuracy of *PPD* with different aggregation methods for EPTs. 100 EPT is used for all models.

The results in Table 8 show the prediction accuracy of *PPD* with two different aggregation methods for EPTs: simple averaging and learned weights. When using learned weights to aggregate logits, the model shows a slight decrease of 0.6%-9.4% in prediction accuracy.

These results suggest that while learned weights provide a more flexible aggregation method, they do not necessarily lead to improved prediction accuracy in this context. The simplicity and stability of the averaging method appear to offer better performance, possibly due to the additional complexity and potential overfitting introduced by learning the weights.

### D.7 MULTI-EXIT ENSEMBLE

While using EPTs for prompt ensemble improves prediction accuracy, it also increases input length, resulting in higher computational overhead and forward pass latency. To address this, we propose the use of a multi-exit ensemble method. In multi-exit ensemble, the hidden states of a prompt token from the last $k$ decoder layers are extracted and averaged to produce the final hidden state, which is then decoded by the decoding head into a guess token, as illustrated in Figure 14. This approach achieves prompt ensemble without the associated computational costs.

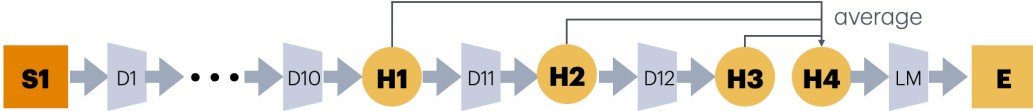

Figure 14: Mult-exit ensemble. 'D1', 'D10', 'D11', and 'D12' are the decoder layers in order. 'S1' is a prompt token and 'H1', 'H2', 'H3' are the corresponding hidden states from the last 3 decoder layers. 'H4' is obtained from averaging these 3 hidden states. The decoding head 'LM' translates 'H4' into a token 'E'.

The hypothesis is that taking the hidden states from the last few decoder layers for ensemble might work because these layers capture increasingly abstract and high-level representations of the input sequence. By averaging the hidden states from multiple layers, we can combine diverse but complementary information, leading to a more robust and accurate final hidden state. Additionally, since the final layers are closest to the output, they are more likely to contain refined and contextually relevant information, making the ensemble more effective.

| Method Name | @1 Top-1 | @1 Top-5 | @2 Top-1 | @2 Top-5 |
|---|---|---|---|---|
| *PPD* without multi-exit | 0.485 | 0.779 | 0.261 | 0.586 |
| *PPD* with 3 exits | 0.422 | 0.723 | 0.214 | 0.517 |
| *PPD* with 2 exits | 0.420 | 0.723 | 0.213 | 0.518 |

Table 9: Prediction Accuracy of *PPD* with and without multi-exit ensemble. 1 EPT is used for all models. $k$ exits refer to the number of exits used.

Table 9 shows the comparison of prediction accuracy of *PPD* with and without mult-exit ensemble. The results indicate that the introduction of multi-exit ensemble with both 2 and 3 exits results in a 7%-18% decrease in prediction accuracy compared to the baseline model without multi-exit.

These findings suggest that the multi-exit ensemble approach, as implemented, does not enhance prediction accuracy and instead leads to a notable decrease in performance. This may be due to the averaging of hidden states from multiple layers introducing noise or reducing the specificity of the representations needed for accurate prediction. Further refinement of the multi-exit ensemble may be necessary to achieve the desired improvements in accuracy.

## E    EFFECT OF BATCH SIZE ON SPEEDUP

| Batch Size | 1 | 2 | 3 | 4 |
|:---:|:---:|:---:|:---:|:---:|
| ***PPD* Speedup Ratio (w/o Tree Attention)** | 1.71 | 1.65 | 1.63 | 1.64 |
| ***PPD* Speedup Ratio (with Tree Attention)** | 2.26 | 1.90 | 1.58 | 1.52 |

Table 10: Speedup ratio of *PPD* compared to baseline across different batch sizes.

As shown in Table 10, consistent speedup ratios are achieved across different batch sizes without tree attention. However, with tree attention, the speedup ratio decreases as batch size increases, a pattern similar to other parallel and speculative decoding methods.

## F    EXPERIMENT DETAILS

For the throughput experiments, each result is obtained by averaging three separate runs. The standard deviations of these runs are reported as error bars in the bar charts. To ensure a fair comparison in our comparative experiments, we maintained consistent hardware settings and software versions.

We selected 3 prompt tokens because adding more would not further increase the expected acceptance length due to the tree size limit. The number of EPTs per prompt token was optimized to maximize throughput.

In Fig. 2, the temperature settings for *PPD*, Eagle (Li et al., 2024a), and Medusa (Cai et al., 2024) follow the default configuration, while the other models use a greedy setting (temperature=0). This choice is based on findings that retrieval-based methods perform significantly worse in non-greedy settings. Similarly, LOOKAHEAD DECODING (Fu et al., 2024), REST (He et al., 2023), and PLD (Saxena, 2023) in Fig. 4 also use a temperature setting of 0 for the same reasons.

## G    LIMITATIONS

Despite its efficiency, we have identified the following limitations of *PPD*:

1. **GPU compute resource constraint.** Since *PPD* trades additional compute resources for increased throughput, its effectiveness depends on the availability of idle GPU compute resources. On a GPU with limited compute resources, the speedup ratios achieved by *PPD* are expected to decrease.

2. **Extended input length.** The improvement in acceptance length with *PPD* is not as significant as the gain in prediction accuracy compared to Medusa. This is because *PPD* must reserve a substantial portion of the input for prompt tokens, which limits the size of the sparse tree that can be used.

## H    SOCIETAL IMPACT

In this paper, we proposed *PPD* to accelerate LLMs easily and cheaply. Since *PPD* reduces the time required for handling a single inference request, it could bring down the cost of deploying LLMs for both the companies and the public. This might lead to increased accessibility of LLM services.

Moreover, latency-sensitive applications like chatbots will benefit greatly from the usage of *PPD* as it reduces the inference latency greatly, thereby enhancing the user experience.

While *PPD* aims to make AI more accessible, there may still be a digital divide where certain communities lack the necessary infrastructure, such as stable internet connections or modern hardware, to fully benefit from these advancements. This could further widen the gap between technology-privileged and underserved populations. On the other hand, *PPD* might be misused by malicious parties to manipulate the output of the original LLM, resulting in the generation of unreliable information and fake data.

