# OpenReview forum: "Hardware-Aware Parallel Prompt Decoding for Memory-Efficient Acceleration of LLM Inference"
_ICLR.cc/2025/Conference — Submitted to ICLR 2025_

### Official Review · Reviewer_3h4G · 2024-10-28

**Soundness:** 3
**Presentation:** 3
**Contribution:** 3
**Rating:** 5
**Confidence:** 3

**Summary:**

The main contributions of the proposed Parallel Prompt Decoding (PPD) method are as follows:

1. PPD introduces prompt tokens that are inserted during the decoding process and achieves a high acceptance rate for long-distance token predictions while preserving output quality. This approach was demonstrated to be parameter-efficient, only requiring tuning a fraction of the parameters.

2. PPD introduces a Hardware-Aware Two-Stage Tree Pruning algorithm, which incorporates a hardware-aware two-stage tree pruning technique that adaptively optimizes the prompt structure of PPD at runtime based on the available compute and memory resources. This allows for efficient deployment across various hardware platforms.

The initial stage is performed offline before the model is deployed. The goal is to reduce the number of prompt tokens in the tree to achieve an optimal tree size. Then the pruning consists of two phases:

1. Candidate Trees Construction: Building trees using only candidate tokens at varying depths, employing algorithms from previous works (e.g., Medusa and Sequoia) to maximize a specific function related to the tree structure.
2. Prompt Tokens Appending: Attaching the maximum allowable prompt tokens to each candidate token from the first step, thereby optimizing the tree structure for better performance.

**Strengths:**

PPD demonstrates substantial speed improvements, achieving up to 2.49× speedup with a minimal runtime memory overhead of just 0.0004%. It also shows potential for synergistic integration with existing speculative decoding methods, leading to further speed enhancements.

**Weaknesses:**

1. As shown in Figure 7b, most of the speedup is from the tree attention technique introduced in Miao 2024. The speedup without it seems to be very moderate.

2. Since the speedup mainly relies on the tree attention part, this casts doubt on the effectiveness of using prompt tokens instead of a more direct approach such as Medusa. While the author claims the method to be hardware-aware, there are no detailed studies on how hardware changes affect the workload and thus affect the results presented in the experiments.

**Questions:**

My concerns are listed above.

---

> ### Author Response · Authors · 2024-11-23
>
> Dear Reviewer 3h4G,
>
> Thanks for your valuable feedback. Here is our response.
>
> Q1: Speedup benefit from tree attention technique and effectiveness of PPD without tree attention as compared to Medusa.
>
> We would like to highlight that almost all speculative/parallel decoding methods benefit significantly from the use of tree attention. As shown in Figure 7b, PPD outperforms all other parallel decoding methods, including Medusa,  without the use of tree attention.
>
> Q2: Effect of hardware changes on workload and speedup ratio
>
> In Figure 5, we present speedup ratios across different GPUs, demonstrating the effectiveness of PPD in adapting to various hardware configurations.
>
> Figures 6b and 6c provide a more detailed analysis of how PPD adjusts the tree size based on the hardware's capabilities. For GPUs with lower computational power, the optimal speedup ratio is achieved with a smaller tree size, as larger tree sizes would incur higher latency overheads, reducing the overall efficiency.

---

### Official Review · Reviewer_F2bw · 2024-10-31

**Soundness:** 2
**Presentation:** 2
**Contribution:** 2
**Rating:** 5
**Confidence:** 3

**Summary:**

This paper presents Hardware-Aware Parallel Prompt Decoding (PPD), a framework designed to accelerate large language model (LLM) inference by leveraging a guess-and-verify process inspired by speculative decoding. PPD introduces a sequence of trained placeholder embeddings, known as prompt tokens, enabling the LLM to generate multiple tokens in parallel within a single forward pass, improving resource utilization and benefiting throughput. These tokens are then verified simultaneously. Compared with changing the model, using pre-trained embeddings incorporates minimal memory and computation costs in both training and inference while achieving a high acceptance rate.

To optimize performance across different hardware, the authors also implement both offline and online tree pruning strategies, ensuring efficient parallel processing on varied platforms. Evaluations on 1.4B, 7B and 13B models demonstrate that PPD delivers up to a 2.49x speedup over standard autoregressive models and a 1.07x improvement over the state-of-the-art Medusa framework, with only half of Medusa's training time requirements.

**Strengths:**

1. PPD effectively uses pre-trained prompt tokens as placeholders, allowing LLM to generate multiple tokens in parallel and significantly boosting throughput.
2. By using prompt tokens instead of changing the model, PPD achieves a state-of-the-art acceptance rate with minimal training overheads.
3. By merging the "Verify" and "Next Guess" steps into a single model execution, PPD improves processing speed and maximizes inference efficiency.
4. PPD incorporates both online and offline decoding tree optimizations, which help maximize performance across various hardware setups.

**Weaknesses:**

1. ***Modest Improvement over SOTA***: While PPD achieves a notable 2.49x speedup over autoregressive models, its 1.07x advantage over Medusa is less impressive. Given that training Medusa is a one-time effort, the slight difference in training time (1.24 vs. 0.52 hours) may not justify the switch for all use cases. Consider the guessing of Medusa only happens on the last stage, **LMHeads**, PDD introduces more decoding workload as each guess is generated through the entire model. This approach may not sustain speedup benefits on larger models or when handling multiple concurrent requests on hardware with high utilization.
2. ***Accuracy on Larger Models is Unclear***: The acceptance length of PPD decreases when moving from Vicuna-7B to Vicuna-13B, and accuracy lags behind Medusa’s. The paper also only presents cumulative accuracy compared with Medusa on 7B model. Further evaluation on larger models (>30B) would help determine if prompt tokens remain effective at greater scales.
3. ***Limited Hardware-Awareness***: While PPD includes hardware-aware tree pruning, the optimization is based solely on forward latency for a single request (batch_size=1). The framework’s effectiveness in more congested or multi-request dynamic environments is unclear and warrants further testing.
4. ***Possibly More Incorrect Guesses***: Though merging "Verify" and "Next Guess" improves hardware utilization, it also means the next guess is based on a set of unverified tokens (guessed in the last cycle). It will increase the distance between an accepted token and next token to guess, and cause additional unnecessary computations as only one of those branches is accepted in the end but PDD has decoded on all the branches.

**Questions:**

1. Could you provide more details on how multi-EPT is trained and used?
2. Could you clarify the x-axis of Fig. 8(a)? It is explained as tree size in 5.4, but decoding 500 branches with just 2.6 average acceptance looks weird.
3. How to combine PDD with speculative decoding and get 1.22x speedup? Did you run draft model on the same GPU? Did you split the verify and next guess into two phases?

---

> ### Author Response · Authors · 2024-11-23
>
> Dear Reviewer F2bw,
>
> Thanks for your detailed and valuable feedback. Here is our response.
>
> Q1.1:  Limited improvement over previous methods.
>
> Our approach has key advantages beyond speedups, like memory saving and cheaper training cost as mentioned in Figure 7a and Table 2\. More free memory enables the use of larger batch sizes, resulting in the potential of improved throughput. Memory efficiency is also vital for long-context tasks, where a large KV cache size could otherwise lead to OOM errors. Additionally, reducing training time significantly lowers training costs, making PPD more accessible for local deployments. Faster training is particularly valuable in online learning scenarios, where the target model distribution may shift over time, requiring frequent retraining.
>
> Q2: Further evaluation on larger models
>
> First, we would like to emphasize that our primary focus, as highlighted in the paper, is on edge and mobile settings. The design of our method, particularly features such as memory savings, is specifically tailored to address the constraints and requirements inherent to these scenarios.
>
> Second, the decrease in acceptance length observed in Table 1 as model sizes increase is attributable to the smaller optimal tree size, rather than a decline in the capability of PPD. In fact, as demonstrated in Figure 6a, the prediction accuracy of PPD improves for larger models due to their greater depth and wider token embeddings, which enhance its overall performance.
>
> Q3: Performance under high GPU utilisation
>
> First, given that our primary use case is the mobile/edge setting (e.g., AIPC), a single-batch scenario is common. In such settings, idle compute resources are often present due to the memory-bound nature of LLM inference, making our method particularly efficient.
>
> Second, in high-utilization scenarios (e.g., large batch sizes), memory availability becomes a critical concern, as highlighted by [FlexGen](https://arxiv.org/pdf/2303.06865). Unlike previous speculative/parallel decoding methods, PPD incurs negligible memory overhead, which helps mitigate these memory constraints and supports efficient inference in such scenarios.
>
> Lastly, we provide analysis of the effect of batch size on the speedup ratio in Appendix E, showing that significant speedups can still be achieved with increased batch sizes. Furthermore, our method is orthogonal to recent advancements in batched speculative decoding [optimization](https://arxiv.org/abs/2406.14066), making it compatible with such approaches.
>
> Q4: Unnecessary computation for incorrect guess tokens.
>
> It is important to note that PPD adopts the same guess-and-verification procedure as previous parallel decoding methods such as Medusa, LookAhead Decoding, and Blockwise Parallel Decoding. Therefore, the additional computation observed is not introduced by our proposed method.
>
> Q5: Details on EPT training and usage.
>
> Please refer to Appendix D for the ablation studies on the design choices of EPT, including the attention mechanism, hyperparameters, and other relevant factors.
>
> In summary, each EPT is associated with its own trainable embedding, which is optimized via backpropagation during training. This ensures an effective initialization that supports accurate multi-token generation. A single prompt token is composed of multiple EPTs. To produce a single output token prediction, the logits generated by these multiple EPTs are averaged, functioning similarly to an ensemble approach. For example, if three EPTs are used per prompt token, the next-next word prediction is obtained by averaging the logits produced by the three EPTs.
>
> Q6: Explanation on x-axis in Figure 8\.
>
> It refers to the tree size as discussed in the paper. The acceptance ratio of 2.6 for a tree size of 500 is reasonable, given that only 3 prompt tokens are used, which effectively bounds the acceptance ratio to 3\.
>
> Q7: How to combine PPD with speculative decoding.
>
> We apply PPD to the draft model to accelerate its draft generation, with the draft model running on the same GPU. In this setup, there exist 2 tiers of guess\&verification processes. The guess-and-verification process is combined for PPD acceleration of the draft mode, while, at the top level, the guess and verification processes are separated between the draft and target models.

---

> > ### Comment · Reviewer_F2bw · 2024-11-24
> >
> > Thanks for clarifying the target scenario.
> > However, I think that makes the project more confusing.
> > 1. The evaluation is tested on A100 and RTX4090 but neither of them is a typical edge devices. For example, Nvidia Orin has high memory capacity (64GB unified memory) with less computation power (64 tensor cores), which will allow you to run large model with single batch size. Even without this concern, the memory saving is not that significant compare with Eagle and Medusa on 7b model.
> > 2. Can you elaborate on Q4? Are all those baselines using a unified guess&verify phase? My concern is that you are guessing on unverified tokens in each inference.

---

> > > ### Author Response · Authors · 2024-12-01
> > >
> > > Dear Reviewer F2bw,
> > >
> > > Thank you for your thoughtful feedback.
> > >
> > > To evaluate the performance of PPD in edge scenarios, we selected models optimized for edge-friendly deployment. The RTX4090 was chosen as a test platform because it is a widely used GPU for personal machines, making it a practical choice for such experiments. Additionally, we conducted tests on the A100 to ensure a fair comparison of speedup against other parallel decoding methods, such as Medusa, which might not be able to be run on GPUs with smaller memory.
> > >
> > > In terms of memory savings, PPD achieved a 41.2% reduction for quantized MobileLLaMA and 23% for quantized Vicuna-7B as shown in Figure 7a, which we believe are significant improvements. Furthermore, unlike Medusa and Eagle, the memory overhead of PPD does not scale with the size of the vocabulary, making it more efficient for larger models and broader applications.
> > >
> > > To clarify more on Q4, we adopted a similar approach to Medusa and other parallel decoding methods which use tree attention. These methods integrate the Guess & Verify phase. Since LLM inference is typically memory-bound, the additional computations do not result in latency overhead. To further optimize computational efficiency, our two-stage tree-pruning algorithm dynamically allocates more computation to tokens associated with higher-likelihood candidates.
> > >
> > > Thank you once again for your valuable time and feedback. If you have any additional questions or concerns, please feel free to reach out, and we will be happy to address them.
> > >
> > >
> > > Best regards,
> > >
> > > Anonymous Authors

---

### Official Review · Reviewer_e6Zd · 2024-11-02

**Soundness:** 2
**Presentation:** 4
**Contribution:** 1
**Rating:** 1
**Confidence:** 4

**Summary:**

This paper looks well written and nicely presented.

Unfortunately its claim for novelty may be challenged, since the main idea of Parallel Prompt Decoding practically overlaps with one in BiTA paper (https://arxiv.org/html/2401.12522v2), published early this year. It uses the same core idea of learnable tokens fed to transformer model to generate speculative continuation. Below is a quote from BiTA paper method description:
"Thanks to the transformer architectures of LLMs, we leverage multiple learnable placeholder tokens known as mask tokens, empowering language models to generate subsequent consecutive future tokens beyond the last input token. During training and inference, the mask tokens are tasked with producing the probability of their next token in corresponding positions."

Now both papers have somewhat different optimisations applied to the core idea. To name a few: BiTA has learnable prefixes, and tree-based decoding. This paper has hardware-aware optimisation.  The speedup range is comparable, with BiTA claiming 2.54x for Vicuna 13B and this paper claiming 2.49x for the same model, though the measurement methodologies differ.

**Strengths:**

Some secondary contributions are deserving positive mention, namely hardware-aware optimisation.

Analysis of accuracy across token positions is helpful in determining the method's performance drivers

**Weaknesses:**

The paper's claim for novelty may be challenged, since the main idea of Parallel Prompt Decoding practically overlaps with one in BiTA paper (https://arxiv.org/html/2401.12522v2), published early this year. It uses the same core idea of learnable tokens fed to transformer model to generate speculative continuation. Below is a quote from BiTA paper method description:
"Thanks to the transformer architectures of LLMs, we leverage multiple learnable placeholder tokens known as mask tokens, empowering language models to generate subsequent consecutive future tokens beyond the last input token. During training and inference, the mask tokens are tasked with producing the probability of their next token in corresponding positions."

Besides an number of weaknesses are probed in the Questions section, namely:
- Lack of proper comparison with Eagle - see Questions section
- Absence of source code (UPD - code found)

**Questions:**

Unlike some other submission, this one has no implementation code.  I wonder if a private code repository could be provided for review purposes?

The paper is not making a complete comparison with SoTA methods like Eagle. Eagle is mentioned in related work and compared in Table 2 on training duration metric. However, for practitioners, the key metric would be the speedup. Extra RAM and training time may often not be the limited factors. Moreover, pretrained Eagle weights could be downloaded for popular models in seconds. See https://github.com/SafeAILab/EAGLE for the download links.

Could you provide results of your method for more modern models, like Qwen and Llama-3? While Vicuna has been a popular testing base for speculative decoding in the past papers, modern practitioners would be more interested in performance of the method with open source models which are in the top of popular benchmarks.

**Details Of Ethics Concerns:**

not sure if this were intentional, but the main idea greatly overlaps with that in BiTA paper (https://arxiv.org/html/2401.12522v2). See Weaknesses section for details.

---

> ### Author Response · Authors · 2024-11-16
>
> Dear reviewer e6Zd,
>
> We thank you for your time and feedback on our submission. Please see our responses below.
>
> > Unlike some other submission, this one has no implementation code.
>
> The reviewer’s comment is untrue. We did submit our code as supplementary material in our original [submission](https://openreview.net/attachment?id=cf7NTWv1iW&name=supplementary_material). Additionally, we also attached the model weights for PPD for the convenience of experiments. Everyone could download the codebase and easily reproduce our results.
>
> > Extra RAM and training time may often not be the limited factors.
>
> In practical applications, RAM usage is a critical factor when deploying AI models on mobile devices and AI PCs with limited memory capacity. Leading companies such as Apple and Samsung are actively pursuing memory-efficient and training-efficient solutions for deploying LLM in their products. While increasing speedup from 2–3 times (as seen with Medusa) to 3–4 times (as seen with Eagle) may or may not enhance the user experience, reducing memory usage to levels compatible with deployment on mobile or IoT devices is the decisive factor that determines whether users can practically utilize LLMs for everyday applications. Therefore, significant memory saving is actually more important than minor speedup improvement.
>
> > It uses the same core idea of learnable tokens fed to transformer model to generate speculative continuation.
>
> We would like to highlight that our main idea is fundamentally different from BiTA in the following ways:
>
> 1. Key Difference-1: Although both papers use the name “prompt tokens”, they are referring to different concepts. Prompt token in BiTA refers to the prompt embeddings appended to each layer as part of the model, which introduces extra computation, memory usage, and complexity in training. In contrast, prompt token in PPD uses the idea of ensemble tokens as part of the input sequence, which decouples the number of trainable parameters from the number of prompt tokens used. Our prompt tokens are virtualized and correspond to arbitrary actual tokens, which increases the acceptance ratio. Note that our approach does not prepend any tokens.
> 2. Key Difference-2: We also propose a novel tree pruning algorithm to minimise the latency overhead introduced by using tree attention with prompt tokens. This is also not mentioned anywhere in the BiTA paper. We do not appreciate the fact that the reviewer claim this is a “ secondary contributions”. The pruning algorithm, as shown in the paper, leads to huge speedup ratio improvement.
> 3. Key Difference-3: We consider the compute and memory capabilities of different GPUs, and propose hardware-aware optimizations to further enhance the flexibility and efficiency of our approaches on diverse hardware.
>
> Still, we thank the reviewer for bringing up the BiTA paper. We acknowledge the good work in this paper. We will cite the paper in the revised version and make comparison with our method, to avoid any similar confusion.
>
> > Lack of proper comparison with Eagle
>
> We presented the speedup comparison here.
>
> |**Temperature**|**Method**| **Speedup Ratio**|
> |-|-|-|
> |**T=1 (Stochastic)**|**PPD**|2.26|
> | |**Eagle**|2.15|
> | |**Sps**|1.38|
> |**T=0 (Greedy)**|**PPD**|2.15|
> | |**Eagle**|2.6|
> | |**Sps**|1.68|
>
> Our proposed method serves as a cost-effective and memory-efficient alternative to speculative decoding, specifically designed for edge and mobile deployments. We emphasize that both memory usage and training time are critical metrics in this context.
>
> ## Memory Efficiency
> | **Model**| **PPD**|**Medusa**|**Eagle**|
> |-|-|-|-|
> |**MobileLLama (4-bit)**|**~0%**|60%|70%|
> |**MobileLLama (16-bit)** |**~0%**|20%|30%|
> |**Vicuna-7b (4-bit)**|**~0%**| 30%| 20%|
>
> ## Training Cost
> | **Method**|**Training TIme**|
> |-|-|
> | **PPD** |0.52 hours|
> | **Medusa**|1.24 hours|
> | **Eagle**|1-2 days|
>
> More free memory enables the use of larger batch sizes, resulting in improved throughput. Memory efficiency is also vital for long-context tasks, where a large KV cache size could otherwise lead to OOM errors. Additionally, reducing training time significantly lowers training costs, making PPD more accessible for local deployments. Faster training is particularly valuable in online learning scenarios, where the target model distribution may shift over time, requiring frequent retraining.
>
> > Could you provide results of your method for more modern models, like Qwen and Llama-3?
>
> It is our intention to compare with SOTA parallel decoding methods like Medusa. Thus, for experiments, we choose the models that most parallel decoding methods also support so that we could make a fair comparison. However, Llama-3 is only released recently and it is hard for us to make a comparison with other parallel decoding methods. If the reviewer insists on requesting this evaluation and provides valid reasons to ensure a fair comparison with other approaches, please do not hesitate to let us know.

---

> > ### Author Response · Authors · 2024-11-19
> >
> > Dear reviewer e6Zd,
> >
> > We would like to sincerely thank you again for your time in reviewing our work.
> >
> > All your concerns have been carefully addressed in our response. Would you mind checking our response and confirming whether you have any further concerns or questions? Any further comments and discussions are welcomed.
> >
> > Best Regards.

---

> > ### Comment · Reviewer_e6Zd · 2024-11-25
> > **insist on withdrawal as this overlaps with BiTA core idea**
> >
> > Dear Authors, thank you for the extensive rebuttal.
> >
> > I admit that I overlooked the implementation code, which I found by now. Sorry about that.
> >
> > The speedup comparison provided by you in the rebuttal can not be put in context as it lacks any information on what these numbers rely to (model, dataset, testing conditions).
> >
> > The amount of similarities with BiTA remains my key concern.
> > You claim that unlike PPD, "Prompt token in BiTA refers to the prompt embeddings appended to each layer as part of the model, which introduces extra computation, memory usage, and complexity in training". As a matter of fact, the core BiTA idea is "trainable mask tokens" which are appended to the input sequence - exactly as it is done in PPD. The prompt tokens are additional idea to enhance the speedup, though as BiTA table 3 shows, most of the effect comes from mask tokens.
> >
> > For the record, I am not related to the authors of BiTA, never even meth them. I continue insisting that the core idea of your paper practically repeats theirs. It is most likely unintentional, as such things happen in science. Yet I strongly suggest to withdraw this paper from this conference and from ArXiv, and repackage it with focus on the tree pruning algorithm which may be a good addition to most other speculative decoding methods. Please ensure that you do proper review of prior work, since I've already seen couple of papers on drafting tree optimization.
> > I am lowering my grade to make the point clear. Thank you for understanding

---

> > > ### Author Response · Authors · 2024-11-26
> > >
> > > Dear Reviewer e6Zd,
> > >
> > > We thank the reviewer’s time in reading our response and admitting the overlooking of our submitted code.
> > >
> > > > As a matter of fact, the core BiTA idea is “trainable mask tokens” which are appended to the input sequence - exactly as it is done in PPD.
> > >
> > > We want to re-emphasize that PPD is fundamentally different from BiTA in the following ways:
> > > 1. **Ensemble Prompt Tokens (EPT)**: In BiTA, a “mask token” corresponds to a **single** token embedding. On the other hand, a “prompt token” in PPD consists of **N** Virtual Tokens, where N is a user-defined value and could be optimized per client. This enables us to explore different further optimizations.
> > > 2. **Ensemble attention masks**: Each ensemble prompt token has its own unique token embedding. The embeddings **within** one EPT are **fully visible** to each other, leading to full attention. Additionally, the attention **between** EPTs is **customized** as shown on line 202 in the paper. This full attention + customized attention enables an **ensemble attention structure**, which is unique compared with any other parallel decoding approaches.
> > > 3. As also acknowledged by reviewers, we proposed a novel **two-stage tree-pruning algorithm** to optimize the run-time computation on different devices.
> > > 4. BiTA uses prefix tokens during training, while PPD does not. We prove the effectiveness of this efficient design in our paper. Also, the training objectives of token embeddings are different. We used a different learning objective involving knowledge distillation than BiTA, which takes in the whole distribution of output tokens into consideration during training.
> > >
> > > In light of the key differences listed above, we do not agree with the reviewer that our main idea overlaps with BiTA. If the reviewer has further questions or opinions on the key differences listed above, please let us know.
> > >
> > > > The speedup comparison provided by you in the rebuttal can not be put in context as it lacks any information on what these numbers rely to (model, dataset, testing conditions).
> > >
> > > We tested on Vicuna-7B with MT Bench. The batch size is 1 and a single A100 is used.
> > >
> > > > as such things happen in science. Yet I strongly suggest to withdraw this paper from this conference and from ArXiv,
> > >
> > > As we explained, our approach is fundamentally different from BiTA. While BiTA also utilizes prompt embedding for multi-token generation, our methodology introduces several unique elements, including “EPT + ensemble attention + two-stage tree pruning + a hardware-aware approach.” These components differentiate our work and contribute novel techniques to the field.
> > >
> > > If the reviewer has further questions regarding our differences from BiTA, please let us know.

---

### Official Review · Reviewer_KH3T · 2024-11-03

**Soundness:** 3
**Presentation:** 3
**Contribution:** 3
**Rating:** 6
**Confidence:** 3

**Summary:**

This paper proposes a parallel decoding algorithm which approximates future outputs in parallel during generation. They leverage multiple prompt tokens which allow for generating outputs at future timesteps. This approach is similar to prefix tuning, where they append trainable prompt tokens to to predict multiple timesteps in advance (although the future prompt tokens are appended at the end of the prompt). They also leverage a dynamic sparse tree search method that allocates a greater number of prompt tokens to more promising candidate sequences. Their method yields comparable results with other parallel decoding methods with fewer trainable parameters / shorter training time. They also outline how their method can be combined with speculative decoding to increase the efficiency gains from this approach.

**Strengths:**

- They present a prompt token-based method to adapt the model to perform parallel tree decoding. These tokens are appended to the end of the sequence and tuned to allow for predicting multiple tokens into the future by approximating tokens generated at future timesteps in order to recover missing conditional dependency information.
- Their hardware-aware sparse algorithm dynamically allocates more or fewer tokens to particular branches depending on their probability, and also incorporates hardware constraints when mapping out the tree
- They show significantly lower training time (and number of trainable parameters) relative to existing approaches like Medusa
- They present clear and comprehensive evaluation of their approach in terms of latency as well as memory usage (including inference efficiency as well as training time and memory considerations)
  - This evaluation also includes ablation of different components of their approach (eg. justifying the benefits of tree attention, as well as of their hardware-aware decoding algorithm)
- They also demonstrate that their approach is complementary with existing speculative decoding methods.

**Weaknesses:**

- The inference-time speedups relative to other prior decoding approaches like Medusa are relatively minor
- Their approach requires fine-tuning, which may make it harder to adapt for different end use cases depending on training availability  (relative to speculative decoding methods)
- The memory savings at inference time of their approach relative to Medusa is smaller for larger models

**Questions:**

None (The methodology and evaluation was very clear)

---

> ### Author Response · Authors · 2024-11-23
>
> Dear Reviewer KH3T,
>
> Thank you for your time and valuable feedback. Please check our responses below.
>
> Q1:  Limited speedup over previous methods.
>
> Our approach has key advantages beyond speedups, like memory saving and cheaper training cost as mentioned in Figure 7a and Table 2\. More free memory enables the use of larger batch sizes, resulting in the potential of improved throughput. Memory efficiency is also vital for long-context tasks, where a large KV cache size could otherwise lead to OOM errors. Additionally, reducing training time significantly lowers training costs, making PPD more accessible for local deployments. Faster training is particularly valuable in online learning scenarios, where the target model distribution may shift over time, requiring frequent retraining.
>
> Q2: The need of fine-tuning for deployment
>
> Thank you for the valuable feedback. We would like to point out that Eagle, a SOTA speculative decoding method, also relies on model-specific features during draft generation. As a result, it encounters similar challenges when maximizing the speedup ratio. PPD reduces the training cost significantly compared to Eagle.
>
> We agree that exploring the transferability of trained PPD tokens across models and enabling training-free deployment are promising directions for future work, and we are excited to investigate these possibilities further.
>
> Q3: The relative memory saving is smaller for larger models.
>
> We would like to emphasize that, as mentioned in the paper, our primary target use case is edge and mobile devices, where memory savings and reduced training costs are critical considerations. In such scenarios, quantized models are commonly employed to further optimize resource efficiency, making the memory saving in the draft model more pronounced.

---

> > ### Comment · Reviewer_KH3T · 2024-11-26
> > **Response**
> >
> > I appreciate the author's response. I will keep my initial score.

---

### Meta-Review · Area_Chair_1o4d · 2024-12-23

**Metareview:**

This paper presents Hardware-Aware Parallel Prompt Decoding (PPD). PPD introduces prompt tokens, enabling the LLM to generate multiple tokens in parallel within a single forward pass. PPD also adopts the idea of tree-structured verification, in which proposed tokens are verified simultaneously, with hardware awareness.

However, 3 out of 4 reviewers raise concerns in:
1. the novelty of this paper is somehow limited. Reviewers find key components of this paper,  e.g.,the use of prompt tokens and parallel verification, sharing ideas with existing papers,
2. 2 reviewers find the evaluation of this paper weak, and the results are not substantial. Baseline methods (published 6 months - 12 months ago) achieved a similar speedup. The models and datasets evaluated in this paper were not quite up-to-date.

**Additional Comments On Reviewer Discussion:**

The author did a good rebuttal trying to address the reviewer's questions, however, on several key topics, e.g., novelty, result substance, the reviewers remained unconvinced.

---

### Decision · Program_Chairs · 2025-01-22

Reject